# An LC–MS/MS Method for the Simultaneous Analysis of 380 Pesticides in Soybeans, Kidney Beans, Black Soybeans, and Mung Beans: The Effect of Bean Grinding on Incurred Residues and Partitioning

**DOI:** 10.3390/foods12244477

**Published:** 2023-12-14

**Authors:** Xiu Yuan, Chang Jo Kim, Hyun Ho Noh

**Affiliations:** Residual Agrochemical Assessment Division, Department of Agro-Food Safety and Crop Protection, National Institute of Agricultural Sciences, Wanju 55365, Republic of Korea; yx0219@korea.kr (X.Y.); rlackdwh1@gmail.com (C.J.K.)

**Keywords:** bean, incurred residue, QuEChERS, liquid chromatography–tandem mass spectrometry, multi-residue

## Abstract

The significance of sample grinding is frequently disregarded during the development of analytical methods, which are often validated with spiked samples that may not accurately reflect incurred residues. This study investigated the particle size of ground beans as a key factor in optimizing extraction efficiency in order to develop a simple quick, easy, cheap, effective, rugged, and safe (QuEChERS)-based modified method for identifying 380 pesticides in beans using liquid chromatography–tandem mass spectrometry. The efficacy of pesticide extraction was found to be significantly affected by particle size. With small particle sizes (>40 mesh), no supernatant was recovered after QuEChERS partitioning. Therefore, a simple modification was performed before partitioning. The modified method was validated for selective extraction of pesticides, limits of quantification, linearity, accuracy, and precision. This method is simple to implement and, therefore, useful for the analysis of pesticide residues in beans.

## 1. Introduction

Pesticides can save approximately 30–40% of agricultural products from the damage caused by various insect pests and diseases [1]. However, pesticide applicators may not always follow pesticide labels or good agricultural practices, resulting in pesticide residues that may exceed the maximum residue limits [2,3,4]. Therefore, it is necessary to systematically monitor pesticide residues. The Korean government implemented a Positive Live System (PLS) for agricultural product quality in January 2019 [5].

The PLS requires the analysis of multiple pesticide residues, necessitating the development of a reliable method for the simultaneous analysis of a large number of pesticides in various food matrices within a short time [6]. The QuEChERS (quick, easy, cheap, effective, rugged, and safe) extraction method, is widely used for the analysis of pesticides in different matrices owing to its cost effectiveness, ease of use, high efficiency, and minimal number of steps [7]. Initially, the procedure involves hand shaking or using a Vortex to extract the homogenized sample with an equal amount of acetonitrile, producing a concentrated final extract without the need for solvent evaporation. Using acetonitrile, as opposed to acetone, offers some advantages. Acetonitrile readily separates from water when the appropriate combination of salts (anhydrous magnesium sulfate, MgSO_4_, and sodium chloride, NaCl) is added, ensuring a well-defined phase separation without the use of hazardous non-polar organic solvents. Additionally, it achieves a high recovery, even for relatively polar pesticides. Anhydrous MgSO_4_ serves as a highly effective drying agent, with its exothermic hydration (around 40 °C) facilitating extraction and confirming the removal of water from the solution without the volatilization or degradation of herbicides. Following centrifugation, which perfectly separates the phases, a rapid procedure called dispersive solid-phase extraction (SPE) is employed for simultaneous clean-up and the removal of residual water. In the method, a primary secondary amine (PSA) sorbent and additional anhydrous MgSO_4_ are mixed with the sample extract. Dispersive SPE, a variation of the SPE methodology, introduces the sorbent to the extract directly without requiring conditioning, allowing for easy clean-up through shaking and centrifugation. PSA, functioning as a weak anion exchanger sorbent, effectively removes fatty acids, sugars, and other matrix co-extractives through hydrogen bonding [8].

QuEChERS has been modified to meet various needs, including dispersive-solid phase extraction (d-SPE) [9], the addition of different amounts of partitioning salts [10], the use of different d-SPE sorbents [11,12], and clean-up with solid phase extraction (SPE) cartridges [6]. QuEChERS has been coupled with liquid chromatography–tandem mass spectrometry (LC–MS/MS) for the simultaneous analysis of pesticide residues [13]. LC–MS/MS in the multiple reaction monitoring mode can distinguish the target compounds from interference with high sensitivity and selectivity. This enables the analysis of a large number of compounds using a simple clean-up step [6,14]. The success of QuEChERS is inseparable from the emergence of MS/MS, and the utility of MS/MS has increased due to the invention of QuEChERS. As the global food trade and the importance of food quality monitoring increase, official international standardization-validated monitoring methods such as the European Committee for Standardization [15], the Association of Official Analytical Chemists [16], and the International or the European Reference Laboratories [17] necessitate miniaturized high-throughput methods. Although there is little uncertainty about the capability of modern technologies and methodologies to efficiently meet analytical requirements, it is crucial to exercise great caution and care to ensure that sample processing is conducted appropriately for meaningful results [18].

Considerable emphasis has been placed on the development and validation of new analytical tools and methods in pesticide research. However, if the selected test samples fail to adequately represent the original batch or unit from which they are derived, the cost, time, and effort involved in using sophisticated analytical instruments and techniques would be ineffective as unreliable and misleading information might be generated [18]. Incurred pesticide residues are pesticide residues in a commodity resulting from the specific use of a pesticide as opposed to residues fortified in laboratory samples [19]. The residues are often difficult to extract because they may be incorporated into cells, starch, or fat particles. As a result, the particle size considerably affects the extraction efficiency of certain pesticide residues [20]. However, sample grinding is often overlooked during the development of analytical methods. A few studies have considered the extractability of residues in cereals [20,21]. It is believed that the increased surface area of the small particle sized samples allows the solvents to access the matrices and thus increases extraction efficiency.

Validation of analytical methods is normally performed through the fortification of samples. However, this process lacks the capacity to demonstrate the efficient extraction of incurred residues. An effective approach is to assess the extraction efficiency of samples bearing incurred residues [22]. Therefore, the aim of the present study was to compare the extraction efficiency of four pesticide-incurred residues in soybeans by particle size to optimize homogenization step, and the same grinding size was applied to kidney beans, black soybeans, and mung beans. In the process of validating the multi-residue method using particle size in the optimized bean samples, a problem in obtaining the supernatant was identified, and the problem was resolved via modification of the sample preparation procedure.

## 2. Materials and Methods

### 2.1. Chemicals and Reagents

Etofenprox 20% EC, azoxystrobin, difenoconazole 28.7 (17.4 + 11.3)% SC, and fludioxonil 20% SC were procured from Gyung Nong (Gyeongsangbuk-do, Republic of Korea), Syngenta (Seoul, Republic of Korea), and Dobang Agro Corporation (Seoul, Republic of Korea), respectively. The pesticide standards in powder form (4, 29, 1, and 5) were purchased from Wako Pure Chemical Industries (Osaka, Japan), Dr. Ehrenstorfer (Augsburg, Germany), Chem Service (West Chester, PA, USA), and Sigma-Aldrich (St. Louis, MO, USA), respectively. All pesticide standard (341) stock solutions (1000 g/mL) were purchased from AccuStandard (New Haven, CT, USA). Acetonitrile (HPLC grade) and formic acid (LC-MS grade) were purchased from Merck (Darmstadt, Germany), and methanol (HPLC grade) was obtained from Fisher Scientific (Incheon, Republic of Korea). Ammonium formate (LC-MS grade) was purchased from Sigma–Aldrich (Darmstadt, Germany). Purified water was obtained using an automatic purification system (AutoMatic Plus GR; WasserLab, Navarra, Spain). All QuEChERS salts (1 g sodium chloride, 4 g magnesium sulfate, 1 g sodium citrate, and 0.5 g disodium citrate sesquihydrate), type 1 d-SPE tubes [150 mg MgSO_4_ and 25 mg primary secondary amine (PSA)], type 2 d-SPE tubes [150 mg MgSO_4_, 25 mg PSA, and 25 mg octadecylsilane (C18)], and type 3 d-SPE tubes [150 mg MgSO_4_, 25 mg PSA, and 2.5 mg graphitized carbon black (GCB)] were purchased from BEKOlut, Bosung Scientific Co., Ltd. (Seoul, Republic of Korea). Organic farming soybeans, kidney beans, black soybeans, and mung beans were purchased from a local market.

### 2.2. Stock Solution Mixture and Matrix-Matched Standard Solutions

Stock solutions of powdered pesticide standards were dissolved in acetonitrile, acetone, or methanol to prepare 1000 μg/mL solutions. Individual stock solutions were combined to prepare a stock solution mixture of 2 µg/mL. The stock solution mixture was diluted with acetonitrile to prepare standard calibration solutions of concentrations 0.2, 0.5, 1, 2, 5, 10, 20, 50, and 100 µg/L. All stock solutions and calibration curve solutions were stored at −20 °C until use. Matrix-matched calibration curve solutions were used for all quantification analyses and were prepared using blank sample extracts following the same procedures used for sample preparation.

### 2.3. Instrument Conditions

A total of 380 pesticide compounds were analyzed using a Shimadzu UHPLC system (Nexera 40 series) coupled with AB SCIEX Triple Quad™ 5500+ (SCIEX, Redwood City, CA, USA). Multiple Reaction Monitoring (MRM) mode was employed for the analysis of the compounds (Appendix A). Compounds in the pesticide mixture were separated at 40 °C using a Kinetex C18 column (2.1 mm × 100 mm, particle size: 2.7 µm) and two mobile phases consisting of (A) 0.1% formic acid and 5 mM ammonium formate in water and (B) 0.1% formic acid and 5 mM ammonium formate in methanol. The 20 min mobile phase gradient program was as follows: 95% A + 5% B for 0–0.2 min, 50% A + 50% B for 0.2–0.5 min, 10% A + 90% B for 0.5–9.5 min, 2% A + 98% B for 9.5–13.5 min, and hold for 13.5–17.0 min, and 95% A + 5% B for 17.0–17.1 min, and hold for 17.1–20 min, respectively, at a flow rate of 0.2 mL/min. ESI-MS was employed in positive (voltage: +5500 V) and negative (voltage: −4500 V) modes with a source temperature of 550 °C and operated in the scheduled MRM mode for quantification analysis. The pressures of the curtain, collision, nebulizer, and drying gases were 35, 10, 50, and 50 psi, respectively.

### 2.4. Contamination of Soybeans and Particle Size Distribution

Soybeans were artificially contaminated by immersion in diluted pesticides registered for use in soybean cultivation in Korea. Azoxystrobin, etofenprox, fludioxonil, and difenoconazole formulations were diluted in 2 L of water to concentrations of 10, 10, 10, and 6.5 μg/mL, respectively, as in a previous study [23]. Two-kilogram batches of soybeans were soaked in pesticide solutions in 5 L plastic beakers for 24 h. The samples were then air-dried under a laboratory hood for 5 days; ground using a high-speed laboratory knife mixer; and sieved through 10, 20, 40, and 60 mesh. The grounded samples were stored at −18 °C until analysis. Untreated soybean, black soybean, kidney bean, and mung bean were also sieved through 10, 20, 40, and 60 mesh.

### 2.5. Sample Preparation Methods

#### 2.5.1. Sample Preparation Method 1

Conventional sample preparation method: 5 g of sample (10–20, 20–40, 40–60, and >60 mesh) was placed in a 50 mL tube, soaked for 30 min in 10 mL of water, and vigorously shaken for 1 min with 10 mL of acetonitrile containing 0.1% formic acid. Thereafter, QuEChERS salt was added, and the mixture was centrifuged at 3500 rpm for 1 min for analysis.

#### 2.5.2. Sample Preparation Method 2

The extraction efficiency of incurred residues in soybeans was compared according to particle size as follows: 10 g of soybean sample for each mesh size (10–20, 20–40, 40–60, and >60 mesh) was placed separately in a 50 mL tube, soaked for 30 min with 20 mL of water, and then shaken vigorously for 1 min with 10 mL of acetonitrile containing 0.1% formic acid. Thereafter, QuEChERS salt was added, and the mixture was centrifuged at 3500 rpm for 1 min. Next, 1 mL of the supernatant was transferred to a d-SPE (150 mg MgSO_4_ and 25 mg PSA) tube, and centrifuged at 12,000 rpm for 5 min. Finally, a 1:1 matrix was matched with acetonitrile before LC–MS/MS analysis.

#### 2.5.3. Sample Preparation Method 3 (Proposed in This Study)

The following simple modification was applied to the conventional method: 5 g of sample (40 mesh) was placed in a 50 mL tube, soaked for 30 min in 10 mL of water, and vigorously shaken for 1 min with 10 mL of acetonitrile containing 0.1% formic acid. The mixture was then centrifuged at 3500 rpm for 1 min, and the supernatant was transferred to another 50 mL tube before adding the QuEChERS salt. Next, the solvent and QuEChERS salt mixture was vortexed for 5 s, centrifuged at 3500 rpm for 5 min, and 1 mL of the supernatant was transferred to d-SPE tubes (type 2: soybean and mung bean, type 3: kidney bean and black soybean) and centrifuged at 12,000 rpm for 5 min. Finally, a 1:1 matrix was matched with acetonitrile before LC–MS/MS analysis.

### 2.6. Method Validation

The developed method (sample preparation method 3) was validated for selective analyte extraction, limits of quantification (LOQ), linearity, accuracy, precision, and matrix effect. For the selective extraction of the analytes, blank soybean, kidney bean, black soybean, and mung bean sample extracts were tested. The LOQ were set at a signal-to-noise ratio higher than 10, satisfying the PLS level, and were validated with sufficient recovery and precision. The linearity of the matrix-matched calibration curve was evaluated as the coefficient of determination (R^2^). The accuracy and precision of the recovery tests were determined in triplicates at fortification levels of 0.001 (0.2 μg/mL spiked to 25 μL), 0.025 (0.5 μg/mL spiked to 25 μL), 0.05 (1 μg/mL spiked to 25 μL), and 0.01 (1 μg/mL spiked to 50 μL), and 0.05 mg/kg (2 μg/mL spiked to 125 μL) in all beans. The linearity, accuracy, and precision were evaluated using the Korean Rural Development Administration criteria (>0.990, 70–120%, relative standard deviation (RSD) ≤ 20%). The matrix effect of all compounds for each bean were calculated as follows [24]:Matrix effect (%) = (slope of the calibration curve obtained using matrix-matched solution/slope of the calibration curve obtained using pure solvent − 1) × 100

## 3. Results and Discussion

### 3.1. Method Development Strategy

Although several studies have described analytical methods for pesticides in soybeans, only a few have described analytical methods for multi-pesticide residues in kidney, black soybean, and mung beans. Table 1 shows published studies [6,25,26,27,28,29,30,31,32,33] on multi-residue sample preparation methods for soybeans. Pesticide analysis involves four main steps: sample grinding, extraction, partitioning, and clean-up [7,34]. Extraction, partitioning, and clean-up have been generally optimized using extraction solvents [35,36], partitioning salts [37], and d-SPE sorbents [38]. However, sample grinding has rarely been evaluated and considered in multi-residue methods. Therefore, we sought to systematically compare sample grinding methods and determine their effect on the extraction efficiencies of the incurred residues. Pesticide contaminated soybeans were obtained by immersion in azoxystrobin, etofenprox, fludioxonil, and difenoconazole formulations dilute solutions. As soybeans absorb water, pesticide residues can enter the soybeans. The incurred pesticide residue in these soybeans is not field generated because the pesticides were not applied during soybean cultivation. Therefore, further investigations involving soybeans cultivated with pesticide treatment in the fields are required.

### 3.2. Effect of Particle Size on the Efficiency of Incurred Residue Extraction from Soybeans

Reducing the sample particle size by grinding is called the comminution process. The purpose of grinding is to reduce the fundamental error and provide a test sample that accurately represents the original sample [18]. To compare the extraction efficiencies for the incurred residues, the contaminated soybean samples were separated by particle mesh size as follows: 10–20-, 20–40-, 40–60-, and >60-mesh soybean particles (>60 mesh indicates the particle size was smaller than 60 mesh). First, the sample extracts were prepared using the widely used sample preparation method 1. However, when the particle size was >40 mesh, the supernatant failed to separate and sample preparation could not be continued (Appendix A). Therefore, method 1 was not considered when comparing the extraction efficiencies. Second, method 2 was used for sample preparation, which was modified from previous studies [6,33]. It was applied for all particle sizes, and the obtained supernatants were used for further processing to prepare the samples. As the pesticide concentrations were high, the extracts were diluted 200-fold using blank extracts to avoid peak saturation and obtain optimal chromatogram. Data were subjected to a one-way ANOVA (*p* < 0.05) using IBM SPSS Statistics 25 (IBM Corp., Armonk, NY, USA) (Table 2). A similar extraction efficiency pattern was observed for all four compounds; the particle size of the 10–20 mesh sample exhibited the lowest extraction efficiency which was significantly lower than that of the >20 mesh samples. For >20 mesh samples, the differences in extraction efficiencies were negligible, and these results are in line with the criteria recommended by the SANTE guideline (1 mm, 18 mesh) for the particle size of low-moisture commodities [39].

For azoxystrobin, fludioxonil, and etofenprox, the average extraction yields improved when the particle size was increased from 10–20 mesh to > 60 mesh. For all target pesticides, the highest extraction efficiency was achieved when the particle size was 40–60 mesh. These findings highlight the importance of validating miniaturized methods with incurred samples, validating sample grinding, and analytical methods [18]. Therefore, a >40-mesh sample was considered suitable to achieve sufficient extraction efficiencies for pesticide residue analysis in soybean.

### 3.3. Effect of Particle Size of All Beans on Partitioning

Very little supernatant was generated from 20–40-mesh samples of kidney bean, black soybean, and mung bean using sample preparation method 1 (Appendix A). It was confirmed that the supernatant volume varied depending on the particle size, and the smaller beans absorbed more solvent when mixed with water, acetonitrile, and MgSO_4_. For sample preparation using method 2, the volume of water was increased compared with that used in previous studies because we found that the smaller particle sizes of ground beans have larger volumes, and a 6 or 10 mL volume was not sufficient to soak the entire 10 g of >20-mesh bean sample [6,33]. However, 20 mL of water was found sufficient to soak the 40-mesh bean samples and obtain an adequate amount of supernatant (Appendix A). However, solvent overflow from the tube is a concern when QuEChERS salts are added owing to the large volume of the sample, water, and solvent. In addition, a large amount of lipids was extracted with acetonitrile when 10 g of sample was used compared with 5 g of sample [7]. According to EN 15662:2009, for dry samples, such as cereals and soybeans, homogenized portion of 5 g ± 0.05 g is recommended for pesticide analysis [40]. Based on the findings, sample preparation method 2, was found to be suitable to compare the extraction efficiencies of incurred residues in different soybean ground sizes but needs some modifications for routine use.

### 3.4. Simple Sample Preparation

We considered the advantages of previous studies involving the QuEChERS-based modification of sample preparation process for beans, such as shaking twice [41], adding different amounts of partitioning salts [42], purification with SPE [6], and purification with new d-SPE sorbents [43]. However, these modifications could not mitigate the supernatant separation failure caused by the small particle size. Soaking the samples in water and extraction with acetonitrile are essential for extraction of pesticides. Therefore, the extraction step before adding the QuEChERS salts was modified. The first improvement was attempted by transferring the supernatant to another tube after shaking, but that many bean particles were observed in the solvents, and no supernatant was available to continue with the purification. When centrifugation was applied after shaking, clear solvents could be obtained without bean matrices. Upon adding the partitioning salts before centrifugation, the acetonitrile layer was completely separated from the water layer (Figure 1).

Consequently, a simple partitioning step was included for extracting pesticide residues from small particle sizes (>40 mesh) of beans successfully. In addition, this method can contribute to enhance the extraction from inconsistent supernatants obtained from dry samples using the conventional method. Finally, simple d-SPE purification efficiency was compared using the d-SPE sorbents (types 1, 2, and 3) for each bean in a simple recovery test for one fortification level (10 mg/kg, *n* = 3) and quantified by 0.5–10, 0.5–25, or 1–25 µg/L calibration curve. The main components in the matrices in soybeans, kidney beans, black soybeans, and mung beans are proteins, carbohydrates, and lipids. which can be effectively removed from various matrices by using PSA. Therefore, PSA-based d-SPE sorbents purification was performed. Figure 2 shows the number of compounds that satisfied the recovery criteria (70–120%). Significant differences were not observed in the number of pesticides, except for type 1 pesticides in soybeans and kidney beans. The d-SPE sorbent type which exhibited satisfactory the recovery rate requirements for majority of the compounds was selected. The final method is summarized in sample preparation (method 3).

### 3.5. Method Validation

A total of 380 pesticides were validated after method development. No interference with the target analytes was observed in the bean samples. The analytes from beans exhibited 0.1–25, 0.1–50, 0.25–25, 0.25–50, 0.5–25, 0.5–50, 1–25, 1–50, or 2.5–50 linear ranges, and coefficients of determination were >0.990. The LOQ of the analytes in the bean samples were set at 1, 2.5, 5, or 10 μg/kg, which were lower than the maximum residue limit and PLS levels, with an S/N ratio > 10. The accuracies and precisions of 284, 331, 301, and 298 out of the 380 residual pesticides in soybeans, kidney beans, black soybeans, and mung beans satisfied the criteria set by the Rural Development Administration (70–120% and RSD ≤ 20% respectively). The method validation data showed that 74.7%, 87.1%, 79.2%, and 78.4% of the 380 pesticides were satisfactorily analyzed as pesticide residues simultaneously in soybeans, kidney beans, black soybeans, and mung beans, respectively. The matrix effect range for most compounds was −50–20%. The method validation results are summarized in Table 3 and detailed results are listed in Appendix A.

## 4. Conclusions

In this study, an LC-MS/MS QuEChERS-based method was developed for the simultaneous analysis of 380 pesticides in beans. The extraction efficiencies of pesticides from beans revealed the significance of the particle size in pesticide residue analysis. Sample preparation method 1 exhibited partitioning problem when bean samples of small particle size (>40 mesh) were used. This problem was resolved through a modification of the extraction procedure. The ease of implementation of this method is expected to widen its usefulness in the analysis of pesticide residues in beans.

## Figures and Tables

**Figure 1 foods-12-04477-f001:**
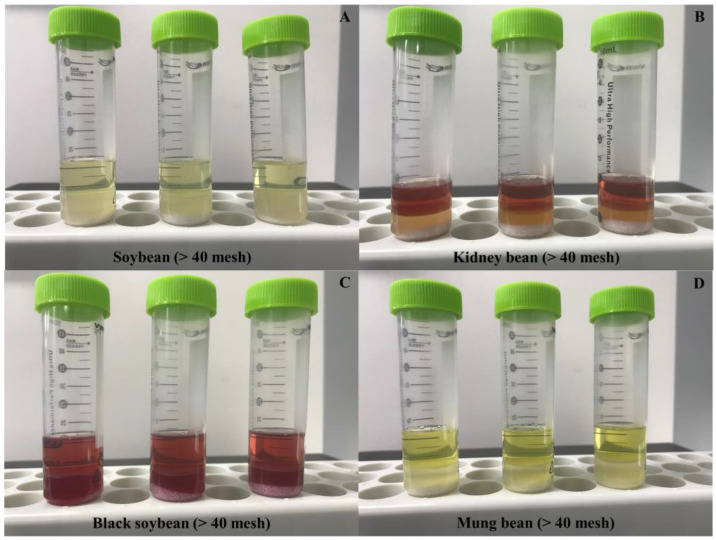
Amount of supernatant obtained using sample preparation method 3 from soybean, kidney bean, black soybean, and mung bean after QuEChERS partitioning. (**A**) soybean, (**B**) kidney bean, (**C**) black soybean, and (**D**) mung bean >40 mesh samples.

**Figure 2 foods-12-04477-f002:**
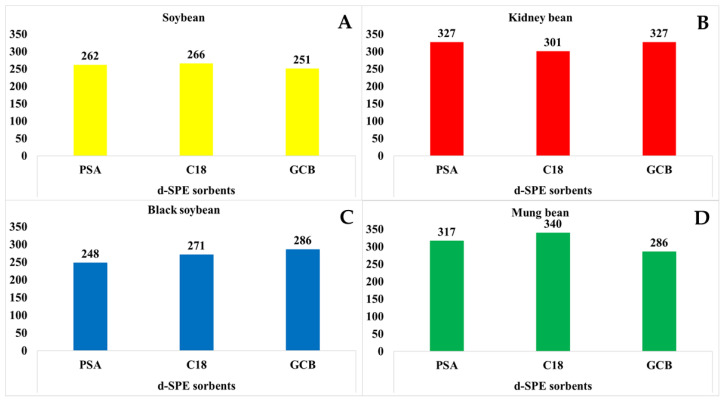
Comparison of the number of pesticide compounds that satisfied recovery criteria using sample preparation method 3 after purification with different sorbents in >40 mesh samples of soybean, kidney bean, black soybean, and mung bean. (**A**) soybean, (**B**) kidney bean, (**C**) black soybean, and (**D**) mung bean.

**Table 1 foods-12-04477-t001:** Analytical method in previous published studies on soybean.

Author	Matrix	Pesticides	Sample	Soak	Extraction	Clean-Up
Bo Tang, 2005 [25]	Kidney bean	9 Organophophorus pesticides	50 g	50 mL water	100 mL acetone	Water dichloromethane partitioning
Satoshi Takari, 2008 [26]	Soybean	76 pesticides	5 g	5 mL water	20 mL acetonitrile	C18 column (Supelclean ENVI-18)
Steven J. Lehotay, 2010 [27]	Soybean	14 pesticides	15 g	13 mL water	15 mL acetonitrile containing 1% acetic acid	d-SPE (MgSO_4_, PSA, and C18)
Rajendra Prasad, 2013 [28]	Soybean	7 carbamate pesticides	5 g	-	10 mL acetonitrile, 5 times	-
Jun Xu, 2015 [29]	Soybean	7 herbicides	10 g	5 mL water	10 mL acetonitrile containing 2% formic acid	d-SPE (MgSO_4_ and C18)
Yongho Shin, 2018 [6]	Soybean	203 pesticides	10 g	6 mL	20 mL acetonitrile	SPE (florisil)
Fernanda Uczay, 2021 [30]	Soybean	5 avermectin pesticides	5 g	10 mL water	10 mL acetonitrile and isopropanol mixture	d-SPE (MgSO_4_, PSA, and C18)
Joseph H. Y. Galani, 2022 [31]	Soybean Kidney bean	99 pesticides	5 g	5 mL water	15 mL acetonitrile	d-SPE (MgSO_4_, PSA, and C18)
Kunming Zheng, 2023 [32]	Soybean	4 pesticides	2 g	10 mL water	10 mL acetonitrile containing 1% acetic acid	d-SPE (MgSO_4_, PSA, and C18)
Arnab Goon, 2022 [33]	Soybean	Spinetoram and its metabolites	10 g	10 mL water	10 mL ethyl acetate and cyclohexane mixture	d-SPE (MgSO_4_, PSA, C18, and GCB)

**Table 2 foods-12-04477-t002:** Average extracted pesticide residue concentrations by particle size.

Crop	Pesticide	Extracted Incurred Residue by Soybean Particle Size Using Method 2 (mg/kg, *n* = 3)			
		10–20 Mesh	20–40 Mesh	40–60 Mesh	>60 Mesh
Soybean	Azoxystrobin	4.53 b ± 0.26	7.33 a ± 0.38	7.40 a ± 0.65	8.10 a ± 0.53
	Fludioxonil	6.33 b ± 3.86	14.5 a ± 0.63	15.60 a ± 4.51	19.10 a ± 1.48
	Etofenprox	0.87 c ± 0.37	2.30 b ± 0.48	3.17 ab ± 0.72	3.73 a ± 0.34
	Difenoconazole	1.00 b ± 0.24	2.30 a ± 0.37	2.30 a ± 0.45	1.20 b ± 0.09

Note: The same lowercase letters across rows indicate no significant differences among the residue concentrations of the same pesticide (*p* < 0.05). Extraction efficiency: a > b > c.

**Table 3 foods-12-04477-t003:** Summary of method validation.

Validation	Criteria	The Numbers of Pesticides (Percentage, %)			
		Soybean	Kidney Bean	Black Soybean	Mung Bean
LOQ	1 μg/kg	15 (5.3%)	57 (17.2%)	11 (3.7%)	29 (9.7%)
	2.5 μg/kg	59 (20.8%)	122 (36.9%)	46 (15.3%)	108 (36.3%)
	5 μg/kg	120 (42.2%)	87 (26.3%)	97 (32.2%)	93 (31.2%)
	10 μg/kg	90 (31.7%)	65 (19.6%)	147 (48.8%)	68 (22.8%)
Linearity	0.1–25 µg/L	80 (28.2%)	299 (90.3%)	45 (48.8%)	10 (3.4%)
	0.1–50 µg/L	170 (59.9%)	0 (0%)	183 (15.0%)	270 (90.6%)
	0.25–25 µg/L	3 (1.1%)	4 (1.2%)	14 (60.8%)	0 (0%)
	0.25–50 µg/L	16 (5.6%)	10 (3.0%)	36 (4.7%)	5 (1.7%)
	0.5–25 µg/L	3 (1.1%)	9 (2.7%)	6 (2.0%)	0 (0%)
	0.5–50 µg/L	6 (2.1%)	1 (0.3%)	5 (1.7%)	10 (3.4%)
	1–25 µg/L	2 (0.7%)	4 (1.2%)	0 (0%)	0 (0%)
	1–50 µg/L	3 (1.1%)	2 (0.6%)	9 (3.0%)	3 (1.0%)
	2.5–50 µg/L	1 (0.4%)	2 (0.6%)	3 (1.0%)	0 (0%)
Matrix effect	<−50%	7 (2.5%)	0 (0%)	1 (0.3%)	0 (0%)
	−50% to −20%	192 (67.6%)	27 (8.2%)	138 (45.8%)	154 (51.7%)
	−20% to 20%	75 (26.4%)	291 (87.9%)	157 (52.2%)	143 (48.0%)
	20% to 50%	8 (2.8%)	12 (3.6%)	3 (1.0%)	0 (0%)
	>50%	2 (0.7%)	1 (0.3%)	2 (0.7%)	1 (0.3%)

## Data Availability

Data is contained within the article.

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
