# Peer review of "An LC–MS/MS Method for the Simultaneous Analysis of 380 Pesticides in Soybeans, Kidney Beans, Black Soybeans, and Mung Beans: The Effect of Bean Grinding on Incurred Residues and Partitioning"

_foods, 2023, doi:10.3390/foods12244477_

Round 1
Reviewer 1 Report
Comments and Suggestions for Authors
The paper reports a LC-MS/MS method for pesticide analysis in beans and soyabean. The paper is very confusing as many experiments were conducted without a clear purpose. Some experiments were done with soyabean and some with different beans, with not clear reason for that. Soyabean and bean are different commodities, with soyabean having a much higher lipid content. There are some missing information on method validation. Some comments are bellow:
Title: the title mentioned beans, but should include soybean. The word simple should be deleted, as the method is not simpler than any other
Line 19 – the information is not correct - Higher mesh numbers = smaller particle sizes
Introduction
Line 33 – not manage pesticide residues, but monitoring them
Line 38/39 – the meaning of QuEChERS should be in parenthesis.
Lines 46-52 – not necessary
Lines 59/60 – not necessary
Lines 67/68 – this statement is not correct, check the JMPR reports
Lines 69/70 – not necessary
Lines 71-75 – the sentences should be replaced by stating clearly what would be investigated in the study and in which commodities
Materials and methods
Line 109– All the MS/MS parameters should be included in the paper: retention time, precursor ion (positive or negative mode), m/z, product ions (qualitative and quantitative), as well as the declustering Potential; Collision Energy; and Collision Cell Exit Potential.
Line 125 – first time soybean appears here. Was it dry soyabean? What is the purpose of different concentrations if the objective is to investigate sample size? Then untreated bean samples were used. What for?
Lines 137-140 – the d-SPE clean-up was not applied here?
Line 143 – what was the solution concentration used to this experiment?
Lines 159/160 – kidney bean and soyabean are dried (pulses) while the other beans are fresh (legumes). How come the water volume is the same?
Lines 164-173 – How the samples were fortified? LOQ is defined as the lowest level at which the method was validated, which in this study was 0.01 mg/kg
Results and discussion
Table 1: all studies used LC-MS/MS? The title should include soyabean. The references are not correctly cited here.
Figure 1 is confusing and not necessary
Lines 207/208 – what is the fundamental error? What is the original samples?
Line 212 – what is the widely used conventional method? In method 1 only 20 and 40 mesh was tested, not what is this conclusion about > 40 mesh comes from? So, finer particles were not good for extraction???? This contradicts the information on line 242 and Table 2, which should contain the standard deviation
Lines 217/218 – what was the concentration?
Line 183 – what was the simple d-SPE purification? In lines 159/160 you mentioned one type of d-SPE for each commodity.
Line 284/285 – this level is very high and with no replicates, the result is useless.
Line 297 – matrix effect experiment was not conducted or shown to indicate whether there is matrix effects.
Figure 3. it is not clear why there are recovery values using different sorbents for all commodities, if line 158-160 specifies one sorbent type for each commodity. Tables S2-S5 does not indicate which sorbent was used
Line 298/299 – LOQ should be set in the commodity, which is 0.01 mg/kg, the lowest level validated
Reviewer 2 Report
Comments and Suggestions for Authors
The topic of this manuscript is of interest to the readers involved with food safety and the Authors paid a lot of attention to the quality of their paper. Their research is design to develop a modified methodology of sample preparation to enable effective, multi-residual analysis of pesticides is beans. Following my analysis of the main text of the manuscript I am sure the paper is worth consideration;
My comments to the Authors are as follows:
1. QuEChERS – please give some more explanation on this methodology, as not everyone is familiar with it
2. Please give some more details on LC/MS detection – what ions exactly were measured? What were the retention times? – in Supplementary Materials
3. Line 33 – „detect and quantify” would be better than ”manage” (a minor point, but I consider this statement misleading).
Reviewer 3 Report
Comments and Suggestions for Authors
The paper is interesting for the developed method and for the impact on the analytical results for the pesticide residue by QuEChERS.
Anyway, the authors should implement the following part:
par 2.2 line 100 - Check the concentration usually the stock solution is 1000 mcg/mL not 1000 g/mL.
Par 3.5 The authors shoul describe better the method validation. In particular they should give information on the spiked levels, the repetition for each fortification level, Standard deviation and CV% parameters. Indicate how many matrices they tested the validation and the information on quantify and Qualifier ions for each pesticides studied by LC/MS/MS. They can use a table to give all information on recoveries, SD, CV% for each fortification level, and a table to give information on the qualifier and quantifier ions with the information on the DP, CP and the other parameters used for the analyssis by LC/MS/MS for each pesticide.
The LoQ shoudl be given for each pesticide analysed.
The Linearity should be described better in the validation paragraph (concentration range for each pesticide and R2 for each pesticide)
The authors gives some of these information as supplementary material anyway it should be better insert the tables in the text.
Round 2
Reviewer 1 Report
Comments and Suggestions for Authors
No additional comments
Reviewer 2 Report
Comments and Suggestions for Authors
I am satisfied regarding the corrections I have recommended.
Comments on the Quality of English LanguageThe paper is easy to follow - no serious issues detected.